# Does a Religious Atmosphere Impact Corporate Social Responsibility? A Comparative Study between Taoist and Buddhist Dominated Atmospheres

**Jing Shao [1], Tianzi Zhang [1], Young-Chan Lee [2]** 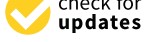 **and Yingbo Xu [1,*]**

1   Business School, Qingdao University, Qingdao 266071, China
2   Department of Business Administration, College of Management and Economics, Dongguk University, Gyeongju 38066, Republic of Korea
*   Correspondence: 2020026557@qdu.edu.cn

**Abstract:** The impact of religion on business has attracted cross-academic attention in recent years. Does the religious atmosphere impact corporate social responsibility (CSR)? This study addressed this question using a sample of Chinese-listed companies from 2010 to 2018. Our findings reveal that firms in regions with a Taoist-dominated religious atmosphere are more charitable and less environmentally invested. In contrast, firms with a Buddhist-dominated religious atmosphere are more ecologically engaged and less charitable. This study extends the literature on the impact of the informal institutional environment on corporate social responsibility by distinguishing the heterogeneity of the impact of Buddhist- and Taoist-dominated religious atmospheres on CSR. It also provides a new perspective for enterprises to formulate corporate social responsibility strategies based on the regional cultural environment. And it also enriches the application of informal institutional theory to the fields of management and religion.

**Keywords:** religious atmosphere; corporate social responsibility; corporate philanthropy; environmental investment; Buddhism; Taoism



## 1. Introduction

The impact of religion on business has attracted cross-academic attention in recent years. A growing number of domestic and foreign scholars have focused on the intersection of religion, culture, politics, economics, and management in the academic field. Many scholars have examined its impact on Western economic growth, business ethics, managerial behavior, and business decisions (Barro 2003; Longenecker et al. 2004; McGuire et al. 2012). Some have also examined the relationship between religion and corporate social responsibility (CSR), focusing on religion and religious atmosphere influencing corporate culture and behavior, among other factors (Díez-Esteban et al. 2017; Hunjra et al. 2021; Tsendsuren et al. 2021). In addition, research on the influence of religion on CSR has begun to attract the attention of scholars (Du 2017; Griffin and Sun 2018; Zolotoy et al. 2019; Xu and Ma 2022). Extant literature indicates that CSR is compatible with the teachings of most religions, as most religions guide believers to do good deeds and emphasize their value (Angelidis and Ibrahim 2004; Brammer et al. 2007). However, most of this literature is based on Western religious beliefs, typically Christianity and Catholicism. Few studies have explored the influence of Eastern religions on CSR in the context of Eastern countries (Barron and Chou 2017; Vu 2018; Su 2019; Suriyankietkaew and Kantamara 2019). As the Chinese government and the private sector have focused more on and demanded CSR fulfillment in recent years, studies on CSR in China have increased. However, there are still gaps in the literature on the religious informal institutional environment. Religions have unique doctrines and influences on their followers; the different implications of these strategic choices on CSR are still unknown.

In forming modern business ethics, informal institutions play an important role in devising official policies (Bampatsou et al. 2017; Elbousserghini et al. 2019; Tolmie et al. 2020). In this informal system, ideology is at the core because it encompasses ethical norms, moral values, and customs. (Estrin and Prevezer 2011; Williamson and Kerekes 2011; Azari and Smith 2012). Religion is a cultural phenomenon that emerged at a certain stage in the development of human society. It has meaningfully shaped culture and is an outstanding social institution. Religious culture is a unique ideology of society, which develops to a particular scale and forms a religious atmosphere. The religious atmosphere as an informal institutional environment refers to the geographic proximity between religions. The degree of influence of religious culture in a region can invariably influence all aspects of people's lives and behaviors (Tarakeshwar et al. 2003; McCullough and Willoughby 2009). We believe that the religious atmosphere significantly influences the fulfillment of corporate social responsibility strategies. This study aims to analyze this environment's effects on local companies' activities. It is based on the idea that religious culture is an integral part of the local social culture and can be used to create an informal environment of religious learning. This study argues that the religious atmosphere significantly influences CSR strategy and that the influence of different religious atmospheres on CSR strategy is heterogeneous, which has not been focused on in previous studies related to Eastern religions and CSR, for example, the difference in the religious atmosphere between Buddhism and Taoism, their differing impacts on CSR strategy, and the difference in CSR for social and natural environments. This study (1) distinguishes between Buddhism and Taoism, which have a long history and extensive influence in China, and (2) examines the impact of the Buddhist- and Taoist-dominated religious atmospheres on philanthropic and environmental investments, respectively.

First, we pursue the religious view of harmony between humans and nature through Buddhism, which may influence the local religious atmosphere, thus creating an informal institutional environment that promotes the protection of the local natural environment. This fact, in turn, may affect the environmental investment behavior of local firms. Second, we argue that temple merit boxes mediate in the charitable agency. This consensus on indirect forms of charity creates an informal institutional environment for a charitable agency. This, in turn, can influence the charitable giving behavior of local firms. Third, we argue that the Taoist view of nature and "inaction", which reveres and follows nature, creates an informal institutional environment that does not destroy or actively change the natural environment. This affects the environmental investment behavior of local firms. Finally, Taoism's religious views of pursuing worldly sentiments and the merit of helping the world create an informal institutional environment of caring for people and giving to society, influencing local firms' charitable behaviors.

Previous studies have explored the impact of religion on CSR, but research on the influence of religion on strategic choices in CSR is scarce. This study primarily considers the degree of geographical proximity between religions or the degree of influence of religious cultures in a region. We start with the premise that religious culture and the local social culture are intertwined and can form an informal institutional environment of a religious nature. In examining the influence of the religious atmosphere on local enterprises' choice of social responsibility strategies, we focus on the heterogeneity of the impact of different religious atmospheres formed by Buddhism and Taoism on CSR strategies. This heterogeneity is ignored in the literature related to Eastern religions and CSR.

We focus on the following research objectives: First, we distinguish the influence of Eastern religions on CSR strategy choices, that is, the impact of the heterogeneity of Buddhist- and Taoist-dominated religious atmospheres on CSR. Through the study and analysis of Buddhist and Taoist teachings, we distinguish the philosophical relationship between these religions and human society and the natural environment. The literature's composition in this field has been enriched by the religious organization of the core literature related to CSR. Second, our study enriches the application of informal institutional theory to management and religion. We explore the impact of the religious atmosphere on how local companies fulfill their social responsibility in the informal institutional en-

vironment formed by Buddhism and Taoism in the region. From a different perspective, it has enriched the research in related fields, such as informal institutional theory and religion. Finally, we added empirical evidence on the impact of religious atmosphere on CSR strategies. Our data analysis systematically demonstrates the differences in the effects of Buddhist and Taoist religious atmospheres on philanthropic and environmental investments.

## 2. Theoretical Background and Hypotheses

### 2.1. Buddhism and Taoism in China

In China, Buddhism and Taoism are traditional religions with a long history of integration with Chinese society and culture, a broad social base, and many followers (Dong 2017). Buddhism and Taoism's ideas profoundly influence individual believers' thoughts and behaviors.

According to Buddhist teachings, there are eight sufferings in life: Life itself is suffering, so one should cultivate it to avoid the suffering of the six degrees of reincarnation in the next life. The only way to be liberated is to look at life and death lightly. Buddha advocates the existence of an afterlife, as this life is predestined. Buddhism advises people to fast, endure, and place their hope in the afterlife. To reach nirvana, Buddhism emphasizes desire as the source of all suffering (Brazier 2003; Du et al. 2014b; Pace 2013). Thus, Buddhists exercise restraint in pursuing materialistic desires, encourage discarding desires, and practice self-discipline (Barnhill 2004). Therefore, its followers are taught to reject materialism (Wiese 2011). Actions with attributes of excessiveness are considered sins of desire in Buddhism. This view profoundly impacts Buddhists, who themselves have a view of material things that extends to themselves and others. Buddhism is concerned with the suffering and impermanence of this world. The belief is that the cause of suffering is selfish desire—greed, envy, and addiction (Ross 1979). Eliminating desire through moderation, calmness, and wisdom can stop suffering. With the disappearance of suffering, there will be absolute peace, tranquility, and perfect bliss, constituting nirvana. The ultimate goal of Buddhism is to save people in this material world and help them attain nirvana.

Chinese Buddhism's moral philosophy, moral code, and moral behavior are embedded in traditional Chinese cultural thought. For example, the Five Precepts and Ten Virtues of Buddhism are not only common to all Buddhists but are also among the most basic moral principles that civilized societies in China must observe. The Five Precepts of Buddhism are consistent with the traditional virtues of benevolence, righteousness, propriety, wisdom, and faith, which Chinese society has always advocated (Liu 2021). The Buddhist Way, with its compassion, equanimity, tolerance, concern for self-reliance, and responsibility—above all, its cosmic view—can be a model for society (Kaza and Kraft 2000).

Taoism is based on Tao as its firmest belief and the ideas of the "Tao Te Ching" as its main doctrine. The Tao advocates respect and virtue and emphasizes human life in harmony with all things. It presents its true nature and keeps its simplicity, unifying the human spirit and form through cultivation. It allows people to clear their minds and return to nature. It advocates kindness, frugality, and non-confrontation with the world. The most fundamental belief of Taoism is the Tao, and all doctrines and teachings are derived from it. Taoism believes that the Tao is all-encompassing, all-pervasive, and the beginning of everything. Taoism fears death, values life, and wants life forever (Hyun 2017). Taoism seeks to cultivate merit to become immortal and emphasizes "both merit and action," more specifically, cultivating the inner work of life and the outer work of helping the world. People cannot become immortal without virtue and good deeds by only refining and nurturing (Kim 2008).

One of the modern implications of Taoist thought is to find a peaceful spiritual harbor for people outside the current materialistic and the fiercely competitive real world. Taoist thought opens up a spiritual world that is much better than the real world to believers and gives them spiritual satisfaction beyond material possession. In addition, Zhuangzi's bold critique of worldly concepts helps people open their minds and break their spiritual

shackles and provides a way to alleviate their misery and stabilize their minds. Taoism rejects the behavior of the strong bullying the weak and the young disrespecting the old. It also advocates helping the world and emphasizes the need for mutual assistance and love among people (Zeng 2006). Taoism's ethical and moral norms, which once played an important role in stabilizing ancient Chinese society and in the harmony of human relationships, are equally valuable in communities.

The survey results show that China has the lowest ratio of religious beliefs to population in the world. However, the survey data indicate that a significant percentage of people in China are religious or have a religious experience and that the number of religious believers in China may be underestimated (Yang 2016). The popular understanding of the religious situation in modern China is that the government supports freedom of religion, but most Chinese people are supporters of atheism. According to the white paper "China's Policies and Practices for Safeguarding Freedom of Religion" published by the State Council Information Office in 2018, China's religious population is increasing yearly, with Buddhism and Taoism accounting for a larger proportion of adherents. Because of their special religious characteristics and liberal forms of belief, ordinary believers have no strict initiation procedure, and it is difficult to count their numbers accurately. Taoism is a traditional Chinese religion and has a certain base of believers. Although Buddhism is an imported religion, it has been closely integrated with traditional Chinese culture and customs due to its early introduction to Chinese society. It has a wide social base and extensive influence. Owing to the nature of Buddhist and Taoist forms of belief, more people are involved in folk beliefs and activities than the official survey figures would suggest. There are currently only about 222,000 Buddhist clergy and 144,000 places of religious activities registered according to the law, including about 33,500 Buddhist monasteries and more than 9000 Taoist palaces. As of 2017, there were 91 religious colleges approved by the State Administration of Religious Affairs, including 41 for Buddhism and 10 for Taoism. The Buddhist Academy of China, High-Level Tibetan Buddhism College of China, Chinese Taoist College, and others are national religious institutions. There are seven national religious groups; the Chinese Buddhist Association and the Chinese Taoist Association are the two most important groups.

Religion has a long history in China. It has been integrated into traditional culture, and to a certain extent, it affects people's consciousness and behavior. However, China is a vast country with great regional differences in culture. There is often a different dominant religion in different geographical regions, which creates a dominant religious atmosphere. Therefore, we can logically infer that these differences have affected the informal institutional environment of regions to a certain extent.

### 2.2. Religious Influence on CSR

As a member of social citizenship, companies have an inescapable responsibility to society and the resource environment (Cui et al. 2015; Yao et al. 2022). As the largest developing country in the world, the number of enterprises in China is increasing yearly. CSR has become the consensus of the Chinese government, enterprises, and society. It is also considered an important contribution to a harmonious society (Lim 2010; Zhou 2006). Many factors influence the fulfillment of CSR. Most religious teachings are consistent with the concept of CSR, as most religions advocate the value of charity (Angelidis and Ibrahim 2004; Brammer et al. 2007). In emerging markets, some well-known determinants that influence CSR, such as ethical culture and corporate governance, are sometimes ineffective in motivating it. Therefore, scholars have turned to informal institutions to seek explanations for CSR behavior (Angelidis and Ibrahim 2004; El Ghoul et al. 2013; Mazereeuw-van der Duijn Schouten et al. 2014).

Weaver and Agle (2002) identified three possible effects of religion on individual behavior: no effect, positive effect, and negative effect. They believed that different influences are based on various religious beliefs. Intrinsic religiosity can positively impact ethics and morality, and religious beliefs can inspire positive attitudes toward social responsibility.

The act of fulfilling social responsibility is based on the attitude towards CSR. The social norms promoted by religion can have a powerful influence that is not limited to believers. To some extent, the social behavior of non-believers can also be improved through the desire for conformity to religious beliefs (Du et al. 2014b; El Ghoul et al. 2013). Several studies have verified that religions in different countries have varying impacts on corporate behavior and CSR (Mazereeuw-van der Duijn Schouten et al. 2014).

Empirical evidence suggests that religion can influence economic choices (Dyreng et al. 2012), and religious belief can influence moral judgment and behavior (Weaver and Agle 2002). El Ghoul et al. (2013) and Hilary and Hui (2009) argue that religious consciousness has a civilizing influence on people's minds. Regardless of individual religious beliefs, once religion is elevated to a social norm, the community's religious beliefs can influence corporate decisions and behavior (Du et al. 2014b; Dyreng et al. 2012; Grullon et al. 2009; McGuire et al. 2012). Kennedy and Lawton (1998) argue that religion acts as an essential social norm that influences individual behavior, promoting social solidarity, providing norms that reduce conflict, and imposing sanctions. Even if directors/managers are not religious, they may (or even must) respond to local religious norms. These religious norms create a general religious atmosphere that interacts with religious employees, customers, and suppliers (Du et al. 2014b; El Ghoul et al. 2013; Kennedy and Lawton 1998).

To some extent, a person's religious self is associated with the place or space where their religion is significant. In sacred places, believers participate in important religious rituals and activities and interact with important religious peers. Gerber et al. (2016) argue that attending religious events leads to greater information sharing and social compliance. Jones-Correa and Leal (2001) found that religion can increase civic and social awareness, which comes from the sense of participation and collaboration prevalent among its believers. In the context of China, we believe that the most important component of the religious atmosphere can be considered in terms of the number of local Buddhist monasteries and Taoist temples.

The religious atmosphere within a community can be seen as a cultural perception that provides a common intellectual framework based on religious ideology (Xia et al. 2021). It also shapes corporate behavior to some extent. Additionally, religion can serve as a social norm influencing CSR behavior (Du et al. 2016). The results of Western religious studies may not suit the context of Eastern religions. This study examines the impact of the religious atmosphere on the social responsibility of local companies in the informal institutional setting of the Chinese context.

### 2.3. Buddhism, Taoism, and CSR

CSR is usually discussed in two parts: corporate philanthropy and environmental responsibility. We are mainly looking at the integration of religion with the local society and culture to form a religiously informal institutional environment, based on which we examine the heterogeneous influence of religious atmosphere on local companies' fulfillment of their philanthropic and environmental responsibilities.

### 2.3.1. Buddhism and CSR

The practice of Buddhist environmental activism takes shape. How is Buddhist practice relevant to the task of caring for the earth? The Buddhist vision of dependent origination, in which everything depends on everything else, can function both as an insight into the nature of reality and as a basis for analyzing environmental problems. Understanding the self from an interdependent ecological perspective radically recasts the task of protecting the planet (Kaza and Kraft 2000).

Buddhist culture contains rich and profound ecological thought. In the Buddhist view, the environment in which human beings live is not mechanical nature, nor is it merely biological nature. It reflects both human moral self-consciousness and the religious practice of "humanizing nature." The environment affects the state of mind, and one must create a

good living environment to transform one's heart. Therefore, many Buddhist theories have ecological implications.

The first is "karmic theory." Buddhism believes that all phenomena are interdependent and interact and that phenomena always have a mutual cause and effect. From an ecological point of view, its significance lies in the warning that individuals, human beings, and society do not exist independently but are closely linked to nature. To damage nature is to damage humankind, thus destroying human existence (Huang and Chen 2009).

Second, "Cosmological schematism" discusses the Chinese Buddhist doctrine of the universe's structure. It focuses on two major issues: the cosmic spatial schema theory and the cosmic concept of time (Fang 2011). From an ecological perspective, it describes the ideal world in people's minds: the orderliness, wholeness, and infinity of the world's ecology emphasizes the importance of the environment to people and expresses humankind's desire to pursue an ideal living environment.

Third, Buddhism regards all things in the world as equal, and that one should not kill for one's selfish desires. This fundamental idea of the right to exist for beings other than humans states that the existence of these creatures is a vital part of the human experience (Huang and Chen 2009).

In addition, Buddhism advocates a spiritually reverent view of everything, believing all things in nature have an aura. Thus, we must protect the natural environment, maintain ecological balance, and reconcile the world. In Buddhism, compassion and wisdom are the qualities that develop out of your practice. In Zen Buddhism, responsibility means responsiveness. Responsiveness is responsibility. A developed, compassionate, loving person influences people unselfconsciously, motivates them, and inspires them to act in similar ways. The whole community benefits (Kaza and Kraft 2000). The Buddhist religious view of pursuing harmony between humans and nature, others, and the universe may influence the local religious atmosphere. Areas with a strong Buddhist culture create an informal institutional environment that promotes the protection of the local natural environment and can affect the environmental investment behavior of local firms. Companies develop more aggressive environmental investment strategies to make their social image more consistent with local ecological views and gain local stakeholders' support. Such a religious atmosphere can positively impact corporate environmental investment behavior. Therefore, we formulate:

**Hypothesis 1.** *A Buddhist-dominated religious atmosphere is positively associated with the environmental investment behavior of local firms.*

The ultimate goal of Buddhists is to achieve nirvana, and Buddhism emphasizes that desire is the source of suffering (Brazier 2003; Du et al. 2014a; Pace 2013). Buddhist followers control the pursuit of material desires through self-discipline (Barnhill 2004). Having too many possessions is considered a sin of desire, so they should focus less on material things (Wiese 2011). The Buddhist tradition of setting up merit boxes in monasteries has three purposes: first, to allow us to cultivate blessings; second, to help in our practice, and third, to help the circulation of the Dharma. Buddhism has three treasures: "Buddha, Dharma, and Sangha." The Dharma represents the Buddhist classics, and the monks represent the monk practitioner. One of the most direct ways to cultivate blessings is to "make offerings to the Three Jewels." The function of the merit box is to receive good money wisely, which is called merit. The monks use good money, give charity, and do good deeds, which is called virtue (Asking Zen with a Mortal Heart 2019). A merit box is a place for good people to do good deeds since many mortals desire to do good. It is also believed that only through merit box donations, offerings to the Buddha, and extensive almsgiving can one achieve merit and success. Monks are more likely to establish donations within monasteries rather than communities. This suggests that areas with strong Buddhist traditions are more likely to have a religious atmosphere. The specific adsorption mechanism of Buddhist monasteries for charitable donations based on merit boxes has the intermediary role of

charitable agents. There is a tendency to make charitable donations through temples in areas with a solid Buddhist atmosphere. Donations of large amounts are made directly through the temple administration and recorded in the temple's merit book. This consensus on indirect forms of social, charitable giving creates an informal institutional environment of a charitable agency, which can influence local businesses' charitable giving behavior. To align their social image with the local view of merit and virtue and to gain the recognition and support of local stakeholders, the tendency is to align corporate charitable giving with the Buddhist tradition of giving. We believe that the religious atmosphere of charitable agencies in Buddhism has a negative impact on corporate charitable giving behavior. Therefore, we formulate:

**Hypothesis 2.** *A Buddhist-dominated religious atmosphere is negatively correlated with the charitable giving behavior of local firms.*

2.3.2. Taoism and CSR

An important foundation of Taoist thought is the natural concept of "the Taoist method of nature." There are two meanings for this. One is that "everything in heaven and earth has a law." That is to say, humans and all natural things follow the universal law, which means "Tao." Another one is that "The essence of the Tao is nature." As a universal law, the Tao is natural, evolves according to its own laws, and is not subject to human intervention. This statement reveals the general law of all things "to follow nature," and the law is universal and cannot be violated. The Taoist view of nature is one of reverence for nature and emphasizes naturalness, spontaneity, simplicity, and freedom from desire (Chan 1963). Taoism emphasizes social order and secular life, focusing on harmonious natural order, tranquility, and transcendent spirituality. Lao Tzu suggests that following Tao results in peace and safe success, while contrary actions lead to conflict and destruction (Maspero 1981). Additionally, "inaction" is one of the most basic spirits of Taoism, a spiritual realm, which is completely consistent with the value orientation of "The Daoist method of nature." The essence of "inaction" is "do nothing, and everything is done." The spiritual state of "inaction" is the direct manifestation of the value of "Taoism and Nature." It is to follow the changes of nature, keep things in their natural state, and not create artificially.

Zhuangzi inherited and developed Laozi's ideas, and "inaction" is usually understood as an attribute of the Tao. Zhuangzi's basic principle is that if we only abide by the Tao, the world will naturally order itself. Zhuangzi viewed the natural world as a single, constant system that is essentially static, in which various species feed on each other to survive. Nature is a constantly fluctuating and self-regulating network in which human existence is neither intrinsic nor permanent. Zhuangzi's ideas further explain why Taoism is not environmentalism (Paul 2005).

Areas with an intensely religious atmosphere will develop a view of revering to and following nature. Emphasis is placed on keeping things natural, spontaneous, and simple rather than artificially altering their basic nature. Therefore, an informal institutional environment that does not artificially destroy or actively change the natural environment will be formed, influencing local enterprises' environmental investment behavior. Companies do not tend to develop strategies for active participation in local ecological investments to align their social image with the local view of nature and gain the support of local stakeholders. We believe that this religious atmosphere can have a negative impact on the environmental investment behavior of firms. Therefore, we formulate:

**Hypothesis 3.** *A Taoist-dominated religious atmosphere is negatively correlated with the environmental investment behavior of local firms.*

Taoist classics contain many noble ideas and concepts, such as charity, to "feed people with joy," and "save the poor and the urgent." Those who wish to live forever must want

to accumulate good deeds and establish merit, be compassionate to things, and forgive themselves and others. The main view of charity in Taoism is the balance between profit and loss. The major point is that nature gives humans equal rights to live and develop, but some people have better opportunities and get rich first. For the affluent, it should be similar to the way of heaven that gives life to all things. As there is no absolute fairness, Taoism advocates the pursuit of a dynamic relative balance to achieve relative justice and harmony. Taoism believes that charity is a conscious "good deed" motivated by inner "compassion" and an instinctive reaction of conscience. Doing good should be a genuine, unselfish spirit of doing for others without expecting anything in return (Hu 2015).

Areas with a solid Taoist atmosphere form an informal institutional environment that advocates helping people, accumulating and performing good deeds, caring for people, and giving to society, influencing local enterprises' charitable giving behavior. Companies tend to develop more aggressive charitable giving strategies to align their social image with the local atmosphere of doing good deeds and giving to the community, as well as to gain the support of local stakeholders. We believe that Taoism's religious view of seeking initiation and merit for the world may influence the local religious atmosphere, positively impacting corporate charitable giving behavior. Therefore, we formulate:

**Hypothesis 4.** *A Taoist-dominated religious atmosphere is positively associated with the charitable giving behavior of local firms.*

### 3. Methodology

*3.1. Sample and Data*

We select the relevant data for analysis to confirm the previously proposed research hypothesis further. Our research object is the philanthropic donation and environmental investment behavior of listed Chinese companies during 2010–2018. China provides a suitable empirical setting to study the impact of the religious atmosphere on corporate social responsibility. First, the central government has issued CSR guidelines since China's 11th Five-Year Plan and initiated the concept of a harmonious society in 2006 (See 2009). Under the impact of traditional culture and government policies, CSR has received significant attention from Chinese firms, especially among listed companies. Second, Buddhism and Taoism are China's two most influential religions (Du 2013). Although economic development has led to positive public attitudes toward wealth distribution, the Chinese are still deeply influenced by traditional values and religious culture (Wang and Qian 2011). This informal system also significantly impacts corporate behavior (Su 2019).

Our research sample comprises all listed companies on the Shanghai Stock Exchange and Shenzhen Stock Exchange from 2010 to 2018. By the end of 2021, the sum market capitalization of the two exchanges reached 91 trillion, ranking second in the world. To test our hypothesis, we combined two major data sources. The first one is the China Stock Market and Accounting Research (CSMAR) database, which is the primary source of financial statements and information for the listed Chinese companies. Therefore, this database is widely used in management and CSR studies (Li et al. 2015; Marquis and Qian 2014). The second data source stems from the Center for the Study of Chinese Religion and Society at Purdue University, where we manually calculated the number of religious sites in each city.

*3.2. Measures*

We define the relevant variables in the following manner to further substantiate the influence of Buddhist- and Taoist-dominated religious atmosphere on the choice of CSR strategy. According to our research hypothesis, the specific variables involved are corporate environmental investment, charitable donations, and regional religious atmosphere.

3.2.1. Corporate Philanthropy

Following previous studies, we measured philanthropic donation (%) as the percentage of total sales given as philanthropic donations in a given year (Gao and Hafsi 2015).

### 3.2.2. Environmental Investment

Because firm size in our study varies significantly across the sample, we used the percentage of environmental investment to total sales to measure the environmental investment of a firm.

### 3.2.3. Regional Religion Environment

The natural logarithm value of the number of Buddhist temples in a given province measured regional Buddhism. Regional Taoism was measured as the natural logarithm value of the number of Taoist temples in a given province.

### 3.2.4. Control Variables

We also included several firm-level and region-level characteristic variables to control the impact of other factors on corporate donation and environmental investment. Firm size was measured as the natural logarithm value of total corporate assets (yuan), because large firms typically attract attention (Edelman 1992) and are susceptible to the impact of stakeholders (Luo et al. 2017). Firm age was measured as the years since the firm was founded. Slack resources significantly influenced corporate social responsibility activities (Seifert et al. 2004). We also included the debt ratio, measured by the percentage of total liabilities as total assets. The natural logarithm value of corporate subsidy income measured subsidy income (yuan). Since ownership is an important source of political legitimacy (Faccio and Lang 2002), state-owned enterprises are less motivated to seek commercial resources through CSR activity (Marquis and Qian 2014). We included state ownership by measuring the proportion of state-owned shares. The political connection was measured by whether a firm's director served as a delegate to the People's Congress or the Chinese People's Political Consultative Conference (Zhang et al. 2016). Regional economic development was measured at the regional level by the natural logarithm value of GDP in a given province (10,000 yuan).

### 3.3. Estimation Methods

To avoid the effects of autocorrelation and heteroscedasticity, the study used a generalized least square method to estimate the religious atmosphere's impact on corporate philanthropy. Furthermore, environmental investment has many zero values in this sample, so we also conducted a Tobit model in this study. The model specification for the hypothesis is estimated as follows:

$$Corporate\,environmental\,investment = \beta_0 + \beta_1 Regional\,Buddhism + \beta_2 Controls + \varepsilon \quad (1)$$

$$Corporate\,philanthropic\,donation = \beta_3 + \beta_4 Regional\,Buddhism + \beta_5 Controls + \varepsilon \quad (2)$$

$$Corporate\,environmental\,investment = \beta_6 + \beta_7 Regional\,Taoism + \beta_8 Controls + \varepsilon \quad (3)$$

$$Corporate\,philanthropic\,donation = \beta_9 + \beta_{10} Regional\,Taoism + \beta_{11} Controls + \varepsilon \quad (4)$$

where $\beta_1$ and $\beta_4$ are the coefficients of the regional Buddhist environment and are used to test Hypothesis 1 and Hypothesis 2; $\beta_7$ and $\beta_{10}$ are the coefficients of the regional Taoist environment to test Hypothesis 3 and Hypothesis 4; Controls represents a collection of control variables; $\varepsilon$ is a random error term.

## 4. Results

### 4.1. Descriptive Statistic and Correlation

Tables 1 and 2 report the descriptive correlation for all the variables in this study. In Table 1, the mean value of regional Buddhism and regional Taoism is 5.525 and 4.015, respectively, and the firms donated a mean value of 4.647 as a natural logarithm value. The correlation between philanthropic donation and regional Buddhism and Taoism is 0.044 and 0.026, respectively. As shown in Table 2, the mean value of environmental investment is 0.01 as a proportion of environmental protection investment to total sales. We also

calculated the variance inflation factor (VIF), finding that the mean of VIF is 2.38, which avoids the risk of multicollinearity.

**Table 1.** Descriptive statistics and correlation for donation dimension.

| Variables | Mean | Sd | (1) | (2) | (3) | (4) | (5) | (6) | (7) | (8) | (9) | (10) |
|---|---|---|---|---|---|---|---|---|---|---|---|---|
| (1) Philanthropic donation | 4.647 | 12.267 | 1.000 | | | | | | | | | |
| (2) Firm size | 22.216 | 1.590 | −0.036 | 1.000 | | | | | | | | |
| (3) Firm age | 15.313 | 5.793 | −0.035 | 0.188 | 1.000 | | | | | | | |
| (4) Debt ratio | 0.429 | 0.227 | −0.116 | 0.537 | 0.238 | 1.000 | | | | | | |
| (5) Subsidy income | 14.552 | 4.954 | −0.006 | 0.029 | −0.159 | −0.027 | 1.000 | | | | | |
| (6) State ownership | 0.047 | 0.141 | −0.004 | 0.162 | −0.029 | 0.051 | 0.013 | 1.000 | | | | |
| (7) Political connection | 0.652 | 0.477 | 0.071 | −0.141 | −0.189 | −0.179 | 0.106 | −0.147 | 1.000 | | | |
| (8) Regional economic development | 10.26 | 0.724 | −0.043 | −0.005 | 0.052 | −0.066 | −0.023 | −0.100 | 0.127 | 1.000 | | |
| (9) Regional Bud | 5.525 | 2.044 | 0.044 | −0.173 | 0.000 | −0.054 | 0.036 | −0.086 | 0.123 | 0.261 | 1.000 | |
| (10) Regional Tao | 4.015 | 2.193 | 0.026 | −0.157 | −0.029 | −0.069 | 0.046 | −0.084 | 0.140 | 0.342 | 0.914 | 1.000 |

Note: N = 6757; correlations greater than |0.03| are significant at the 0.05 level.

**Table 2.** Descriptive statistics and correlation for environmental investment dimension.

| Variables | Mean | Sd | (1) | (2) | (3) | (4) | (5) | (6) | (7) | (8) | (9) | (10) |
|---|---|---|---|---|---|---|---|---|---|---|---|---|
| (1) Environmental investment | 0.010 | 0.783 | 1.000 | | | | | | | | | |
| (2) Firm size | 22.216 | 1.590 | −0.009 | 1.000 | | | | | | | | |
| (3) Firm age | 15.313 | 5.793 | −0.009 | 0.188 | 1.000 | | | | | | | |
| (4) Debt ratio | 0.429 | 0.227 | −0.009 | 0.537 | 0.238 | 1.000 | | | | | | |
| (5) Subsidy income | 14.552 | 4.954 | −0.001 | 0.029 | −0.159 | −0.027 | 1.000 | | | | | |
| (6) State ownership | 0.047 | 0.141 | 0.032 | 0.162 | −0.029 | 0.051 | 0.013 | 1.000 | | | | |
| (7) Political connection | 0.652 | 0.477 | −0.017 | −0.141 | −0.189 | −0.179 | 0.106 | −0.147 | 1.000 | | | |
| (8) Regional economic development | 10.26 | 0.724 | −0.006 | −0.005 | 0.052 | −0.066 | −0.023 | −0.100 | 0.127 | 1.000 | | |
| (9) Regional Bud | 5.525 | 2.044 | 0.009 | −0.173 | 0.000 | −0.054 | 0.036 | −0.086 | 0.123 | 0.261 | 1.000 | |
| (10) Regional Tao | 4.015 | 2.193 | 0.009 | −0.157 | −0.029 | −0.069 | 0.046 | −0.084 | 0.140 | 0.342 | 0.914 | 1.000 |

Note: N = 7066; correlations greater than |0.03| are significant at the 0.05 level.

## 4.2. Hypothesis Testing

Model 1 includes control variables. Models 2 and 3 add regional Buddhist and Taoist environments as independent variables, respectively. Tables 3 and 4 report the empirical results of the estimate for corporate donation and environmental investment, respectively.

**Table 3.** Estimation of environmental investment.

| Variables | DV: Environmental Investment | | |
|---|---|---|---|
| | **Model 1** | **Model 2** | **Model 3** |
| Firm size | 0.754 *** | 0.754 *** | 0.754 *** |
| | (0.169) | (0.169) | (0.169) |
| Firm age | 0.024 | 0.024 | 0.024 |
| | (0.035) | (0.035) | (0.035) |
| Debt ratio | −0.519 | −0.519 | −0.519 |
| | (1.055) | (1.055) | (1.055) |
| Subsidy income | 0.149 *** | 0.149 *** | 0.149 *** |
| | (0.042) | (0.042) | (0.042) |
| State ownership | 3.739 * | 3.739 * | 3.739 * |
| | (1.522) | (1.522) | (1.522) |
| Political connection | 0.163 | 0.163 | 0.163 |
| | (0.41) | (0.410) | (0.410) |
| Regional economic development | 10.265 *** | 10.265 *** | 10.265 *** |
| | (1.228) | (1.228) | (1.228) |
| Regional Bud | | 17.978 * | |
| | | (7.544) | |
| Regional Tao | | | −6.171 * |
| | | | (2.59) |

**Table 3.** *Cont.*

| Variables | DV: Environmental Investment | | |
|---|---|---|---|
| | **Model 1** | **Model 2** | **Model 3** |
| Constant | −150.807 | −228.187 | −131.731 |
| | (3838.505) | (3838.671) | (3838.497) |
| Observations | 7066 | 7066 | 7066 |
| Pseudo R$^2$ | 0.180 | 0.180 | 0.180 |
| LR chi2 | 455.20 *** | 455.20 *** | 455.20 *** |
| Log likelihood | −1037.2208 | −1037.2208 | −1037.2208 |
| Industry dummies | yes | yes | yes |
| Province dummies | yes | yes | yes |

Standard errors are in parentheses *** $p < 0.001$, ** $p < 0.01$, * $p < 0.05$.

**Table 4.** Estimation for donation.

| Variables | DV: Philanthropic Donation | | |
|---|---|---|---|
| | **Model 1** | **Model 2** | **Model 3** |
| Firm size | 0.130 *** | 0.130 *** | 0.130 *** |
| | (0.029) | (0.029) | (0.029) |
| Firm age | −0.037 *** | −0.037 *** | −0.037 *** |
| | (0.006) | (0.006) | (0.006) |
| Debt ratio | −2.084 *** | −2.084 *** | −2.084 *** |
| | (0.190) | (0.190) | (0.190) |
| Subsidy income | −0.014 ** | −0.014 ** | −0.014 ** |
| | (0.005) | (0.005) | (0.005) |
| State ownership | −0.346 * | −0.346 * | −0.346 * |
| | (0.170) | (0.170) | (0.170) |
| Political connection | 0.425 *** | 0.425 *** | 0.425 *** |
| | (0.073) | (0.073) | (0.073) |
| Regional economic development | −0.687 *** | −0.687 *** | −0.687 *** |
| | (0.145) | (0.145) | (0.145) |
| Regional Bud | | −4.196 *** | |
| | | (1.111) | |
| Regional Tao | | | 1.440 *** |
| | | | (0.381) |
| Constant | 8.289 *** | 26.35 *** | 3.836 * |
| | (1.681) | (5.355) | (1.866) |
| Observations | 6757 | 6757 | 6757 |
| Wald chi2 | 928.61 *** | 928.61 *** | 928.61 *** |
| Industry dummies | yes | yes | yes |
| Province dummies | yes | yes | yes |

Standard errors are in parentheses *** $p < 0.001$, ** $p < 0.01$, * $p < 0.05$.

Hypothesis 1 predicts a positive effect of the amount of regional Buddhist temples on corporate environmental investment. As shown in Table 3, Model 2, the coefficient of regional Buddhism is positive and significant ($\beta = 17.978, p < 0.05$), suggesting that a religious atmosphere dominated by Buddhism promotes the environmental investment behavior of local firms. As shown in Table 3, Model 3, the coefficient of regional Taoism is negative and significant ($\beta = -6.171, p < 0.05$), which supports Hypothesis 3.

Hypothesis 2 predicts a negative effect of the number of Buddhist temples on corporate philanthropy. As shown in Table 4, Model 2, the coefficient of regional Buddhism is negative and significant ($\beta = -4.196, p < 0.001$), which supports Hypothesis 2. As shown in Table 4, Model 3, the coefficient of regional Tao is positive and significant ($\beta = 1.440, p < 0.001$), suggesting that Taoism's religious atmosphere promotes local firms' philanthropic donations.

## 5. Discussion

Our research data and empirical results suggest that a religious atmosphere can impact the choice of CSR strategies. In this study, our empirical results show that the

Buddhist-dominated religious climate is positively related to the environmental investment behavior of local firms and negatively related to the charitable giving behavior of local firms. The Taoist-dominated religious climate, on the other hand, is negatively related to the environmental investment behavior of local firms and positively related to the charitable giving behavior of local firms. This also confirms that the CSR philosophy is compatible with most religions' teachings because religions typically guide believers to goodness and good work, emphasizing the value of good deeds. As noted earlier, a religious atmosphere created by combining religion and social culture is a regional informal institutional environment. It makes a social consensus that affects people's ideology, words, and deeds within its scope, influencing their perception and evaluation of CSR. Therefore, companies should fully consider the local religious atmosphere and the social consensus formed on it when formulating their social responsibility strategies, even if this influencing factor has not yet been considered. Further, it is crucial to examine the differences in the influence of the dominant religious atmosphere on the choice of philanthropic or environmental investments. This will enable companies to fulfill their social responsibility effectively.

The following theoretical contributions and practical implications are described based on the study's results:

First, we have broadened the literature on the core ideas of Buddhist and Taoist teachings, comparing and analyzing the philosophical relationship between Buddhism, Taoism, the natural environment, and human society. Second, we distinguish the heterogeneity of the impact of Buddhist- and Taoist-dominated religious atmospheres on CSR regarding the informal institutional environment formed locally through combining Buddhism and Taoism with social culture. We also explore the effect of this heterogeneity on the strategic choices of local firms to fulfill their social responsibility, enriching the application of informal institutional theory in management and religion from a unique perspective. Third, we extend the study of religion on CSR strategies by expanding CSR from a single strategy to charity and environmental responsibility. We emphasize the complexity of the impact of different religions on CSR strategy decisions, which enriches the literature in this field.

Despite its relevant contributions, this study has some limitations. First, the only data used are from 2008, which is cross-sectional. Scholars can use updated data or continuous data in the future to verify whether the present results are supported. Second, we measure religious atmosphere based on the number of temples in the region, which is meaningful but does not directly reflect the attitudes of businesses or entrepreneurs toward religion. Scholars can expand the religious atmosphere to include more dimensions and, thus, enrich the study of the religious atmosphere and CSR. Third, our study focuses on the heterogeneous influence of the religious atmosphere dominated by Buddhism and Taoism on CSR strategy choices in the Chinese context. Future research can examine whether there are differences in the impact of the dominant religious atmosphere in the West on CSR in a larger international context. Studying the influence of the religious atmosphere on CSR in a social environment where Eastern and Western religions coexist and have equal influence is possible.

**Author Contributions:** Conceptualization, J.S. and Y.X.; methodology, Y.-C.L. and Y.X.; writing—original draft preparation, J.S. T.Z. and Y.X.; writing—review and editing, J.S. and Y.X.; funding acquisition, J.S. All authors have read and agreed to the published version of the manuscript.

**Funding:** This research was funded by the Research Project of Qingdao University Humanities and Social Sciences Foundation Project Cultivation, grant number No. RZ2100004790.

**Institutional Review Board Statement:** Not applicable.

**Informed Consent Statement:** Not applicable.

**Data Availability Statement:** Not applicable.

**Conflicts of Interest:** The authors declare no conflict of interest.

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
