# Peer review of "Does a Religious Atmosphere Impact Corporate Social Responsibility? A Comparative Study between Taoist and Buddhist Dominated Atmospheres"

_religions, doi:10.3390/rel14010113_

Round 1
Reviewer 1 Report (Previous Reviewer 3)
Dear authors,
As I can see from the peer-reviewed version, you have seriously revised the content of the article. However, I have the following two remarks to the manuscript presented:
1. Table 1 does not provide data on units of measurement for the used variables, for example, in what units you measure Philanthropic donation, Firm size, etc. Please correct this.
2. Please add the Discussion section where you should provide clear explanations for the meaning of calculations done and the contribution of hypotheses to the development of scientific knowledge.
Best wishes.
Author Response
Dear Reviewer,
We would likely to thank you for comments and the commitment to this manuscript. We hope that you find it our revision has improved the quality. Please find out responses below.
1. Table 1 does not provide data on units of measurement for the used variables, for example, in what units you measure Philanthropic donation, Firm size, etc. Please correct this.
Response to comment 1:
Many thanks for pointing out these concerns for the variable measurement. We have labeled the units of the relevant variables in the methodology section. These specific changes have also been made in the revised manuscript in responding to your concerns.
2. Please add the Discussion section where you should provide clear explanations for the meaning of calculations done and the contribution of hypotheses to the development of scientific knowledge.
Response to comment 2:
We sincerely appreciate your valuable input, which we have used to make adjustments to the discussion section.These specific changes have also been made in the revised manuscript in responding to your concerns.
Best wishes.
Reviewer 2 Report (New Reviewer)
Table 1 and Table 2 have the same title. I suggest to indicate in the title what the tables refer to: one to the environmental dimension, the other to the philanthropic dimension.
A discussion needs to be added, because despite the fact that it was indicated in the title in chapter 4, it was not included there. You can combine the discussion with the conclusion in Chapter 5.
Author Response
Dear Reviewer,
We would likely to thank you for comments and the commitment to this manuscript. We hope that you find it our revision has improved the quality. Please find out responses below.
1. Table 1 and Table 2 have the same title. I suggest to indicate in the title what the tables refer to: one to the environmental dimension, the other to the philanthropic dimension.
Response to comment 1:
Many thanks for pointing out these concerns for the variable measurement. We have adjusted the title of Table 1 and 2.
2. A discussion needs to be added, because despite the fact that it was indicated in the title in chapter 4, it was not included there. You can combine the discussion with the conclusion in Chapter 5.
Response to comment 2:
We sincerely appreciate your valuable input, which we have used to make adjustments to the discussion section. These specific changes have also been made in the revised manuscript in responding to your concerns.
Best wishes.
This manuscript is a resubmission of an earlier submission. The following is a list of the peer review reports and author responses from that submission.
Round 1
Reviewer 1 Report
The article is not a groundbreaking one in its field. Please try to raise overall quality and clarity in order to make this contribution more meaningful.
Reviewer 2 Report
This is a very interesting study. However, from the viewpoint of religious studies, the lack of religious knowledge challenges the validity of the research finding.
Line 94-160
Due to the high degree of religious syncretism in the Chinese culture, it has always been contestable in the field of religious studies of how to distinguish Buddhism, Taoism, Chinese popular religion, or the religious hybridity. This section is too simple.
To publish in an academic journal designate for religious studies, the authors must provide more information (such as visual factors, self-identificationt) to show how they determine what is Taoism, what is Buddhism, or, even, why is not religious hybridity.
Line 223-241 : not Buddhist thought, please provide reference. The book, 'Dharma Rain: sources of Buddhist Environmentalism', is a good start.
Line 253-261: please provide reference
Reviewer 3 Report
Dear authors,
Despite the fact that as a reviewer I highly appreciate the research idea and the design of the paper, I hope that the following remarks and suggestions would be useful.
First, you need to state the research objective clearly in the Introduction section and explain its difference from the previous research.
Line 323 Our research objective – please check it (probably object?)
In my opinion, the section 3. Methodology does not explain clearly the connection between the tested hypotheses and the estimation methods. Please enhance this section by adding this information.
Line 385 – Statistic – please check it.
Now the section with discussion has the general conclusions, limitations, etc. which is usually found in the conclusion section. Please enhance this section with the discussion of the results obtained and add the section “Conclusions” (where you can put the text from the lines 417-466).
Best wishes!